# REVISITING TERNARY NEURAL NETWORKS TOWARDS ASYMMETRIC THRESHOLDS AND UNIFORM DISTRIBUTION

## ABSTRACT

Recently, researchers have made significant progress in ternary logic circuits, which has spurred the utilization of Ternary Neural Network (TNN) due to its compatibility with ternary coding instead of the 2-bit coding used in binary system. However, TNN exhibits significant accuracy degradation compared to its full-precision counterpart. Therefore, we are motivated to revisit ternary neural networks and enhance their performance. To fully leverage the limited representation space, we apply a uniform distribution to three quantized values $\{-1, 0, +1\}$ to maximize the information entropy. To balance the representation ability of TNN while considering convenient hardware implementation, we adopt the asymmetric thresholds and symmetric scaling factors quantization scheme and introduce the bi-STE optimization method. Moreover, a two-stage knowledge distillation scheme is employed to further enhance the performance. Experimental results demonstrate the effectiveness of the proposed method for TNNs, achieving a top-1 accuracy of 74.5% for ResNet-50 on ImageNet. This outperforms previous ternary quantization methods by a large margin and even surpasses representative 2-bit quantization methods such as LSQ (73.7%).

## 1 INTRODUCTION

Deep Neural Networks (DNNs) have been widely applied in computer vision applications such as image recognition and object detection *etc*. However, modern neural networks usually require a large amount of storage and computational resources, which impedes their deployment on edge devices. To address this challenge, model compression has become an important technology for applying DNNs on mobile devices with limited storage and computational power. Various model compression methods have been developed, including quantization (Zhou et al., 2016; Choi et al., 2018; Zhang et al., 2018; Esser et al., 2019), pruning (Luo et al., 2017; Neklyudov et al., 2017; Lin et al., 2020), knowledge distillation (Hinton et al., 2015; Xu et al., 2018), compact network design (Howard et al., 2017; Ma et al., 2018; Sandler et al., 2018; Han et al., 2020), *etc*. Among these methods, quantization is an efficient way to compress and accelerate neural networks, since it utilizes fewer bit-widths to represent the weights and activations without altering the network architecture.

In the works of Courbariaux et al. (2015; 2016), the authors proposed to train the DNN with weights and activations constrained to either -1 or +1. Subsequently, researchers extended the representation space to $\{-1, 0, +1\}$ (Li et al., 2016; Mellempudi et al., 2017; Zhu et al., 2017), leading to the emergence of Ternary Neural Network (TNN). However, TNN's performance is limited on the conventional hardware, such as CPUs or GPUs, which typically rely on binary logic circuits. To overcome this hardware limitation, Alemdar et al. (2017) design a bespoke hardware architecture for TNN, which is implemented using FPGAs and ASICs. Despite the promising results of this design, the specialized hardware and equipment costs still restrict the usability of TNN.

Information theory suggests that $e$-nary coding is the most efficient mode, with ternary coding theorized to be more effective than binary coding. The cost function $cost = r * n$ approximates the hardware complexity of an $N$-nary system, where $r$ is the number of different symbols that the system uses and n is the number of digits in a code word. In binary logic circuits, the most basic unit,

an inverter, requires two transistors, resulting in $cost = 2$. In contrast, the costs are more variable depending on the design schemes for the ternary logic circuits (Lin et al., 2009; Huang et al., 2017; Srinivasu & Sridharan, 2017; Jeong et al., 2019). Studies have shown that the cost factor for ternary circuits is close to 3, which indicates that the actual efficiency of ternary systems surpasses that of binary systems. Given this conclusion, we aim to develop a more efficient and high-performance TNN.

In this paper, a novel ternary quantization method is proposed for training and inferring ternary neural networks. Firstly, we identify that the distribution of three quantization candidates is not proportional in conventional ternary quantization methods. To address this issue, a normalization method with a learnable factor is employed to adjust the distribution of three quantization candidates. Secondly, after a comprehensive study, the scheme of asymmetric thresholds and symmetric scaling factors is adopted, which can be easily applied using a *comparer* in hardware. And a novel bi-STE optimization method is proposed, which increases the model's representation ability. Finally, building upon the methodologies of binary neural networks, we utilize a two-stage training scheme to progressively optimize the ternary model. In the first stage, activations are quantized, and then the weights are quantized in the second stage. Experimental results on CIFAR-10 and ImageNet-1k datasets demonstrate the effectiveness of the proposed method.

Our contributions to the optimization of ternary neural networks can be summarized as follows:

- We introduce asymmetric thresholds to enhance the TNN's representation capacity, while symmetric scaling factors facilitate hardware implementation. Additionally, we introduce the generalized bi-STE method to make the optimization process more adaptable.

- We propose a regularization method for ternary neural networks that enforces a uniform distribution among three quantization candidates.

- We employ a two-stage knowledge distillation method to further enhance the performance. Experimental results on the CIFAR-10 and ImageNet-1K datasets demonstrate that the proposed method achieves a better top-1 accuracy compared to state-of-the-art ternary approaches and even outperforms 2-bit quantization methods.

## 2 RELATED WORKS

### 2.1 TERNARY NEURAL NETWORKS

Among quantization methods, ternary quantization has drawn noteworthy attention. Ternary Weight Network (TWN) proposed in Li et al. (2016) is the first approach which can achieve good results on large dataset like ImageNet (Russakovsky et al., 2015). Ternary weights are also investigated in the work of Zhu et al. (2017). Trained Ternary Quantization proposed in Mellempudi et al. (2017) learns both ternary values and ternary assignments. These methods can achieve comparable accuracy with full-precision counterparts on ImageNet. However, only the weights are quantized, leaving the activations in floating-point format. Alemdar et al. (2017) propose to quantize both the weights and activation to ternary. In the work of Wang et al. (2018), the authors also propose to quantize both weights and activations using a two-step procedure, in which the weights and activations are quantized to ternary and 2-bit respectively. The study of ternary neural networks stagnate in recent years due to the in-adaptability of the hardware. However, along with the development of ternary logic circuits, we anticipate the capability to revisit and promote the application of ternary neural networks.

### 2.2 TERNARY LOGIC CIRCUITS

The research of ternary logic circuits dates back to 1980s when the design of silicon CMOS ternary occur as a moderate breakthrough, where one p-MOSFET, two resistors and one n-MOSFET are in-series connected (Miller, 1993). Such circuits simplify the design, but it require two passive resistors and increase the production complexity. Lin et al. (2009) propose a multi-diameter (multi-threshold voltage) CNTFET-based ternary design to implement the ternary logic gates. Moreover, a few ternary arithmetic circuits such as the HA and multiplier have been also designed to decrease the complexity. Srinivasu & Sridharan (2017) represent an algorithm for synthesis that combines

a geometrical representation with unary operators of multi-valued logic. The geometric representation facilitates scanning appropriately to obtain simple sum-of-products expressions in terms of unary operators. Jeong et al. (2019) propose a wafer-scale ternary CMOS technology based on a single threshold voltage and relies on a third voltage state created using an off-state constant current that originates from quantum-mechanical band-to-band tunneling. In 2020, Kim et al. (2020) propose a logic synthesis methodology with a low-power circuit structure for ternary logic, which uses the body effect to mitigate the excessive power consumption for the third logic value. Energy-efficient ternary logic circuits are designed with a combination of synthesized low-power ternary logic gates and the proposed methodology is applicable to both unbalanced $\{0, 1, 2\}$ and balanced $\{-1, 0, 1\}$ ternary logic. The progress in ternary logic circuits motivates us to revisit TNN and boost its performance.

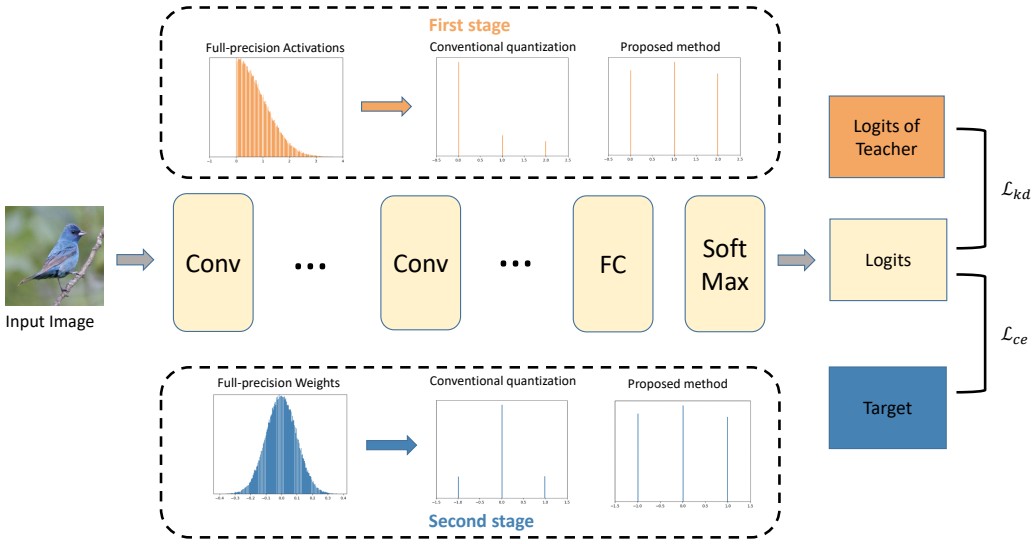

Figure 1: Illustration of the proposed method.

## 3 APPROACH

In this section, we first briefly introduce the formulation and objective of ternary neural networks in Sec. 3.1 . Then the proposed scheme is presented with three components: uniform distribution for weights and activations in Sec. 3.3 , bi-STE optimization method for asymmetric thresholds in Sec. 3.2 , a two-stage knowledge distillation in Sec. 3.4 .

### 3.1 PRELIMINARIES

An arbitrary neural network with $L$ layers has $L$ sets of weights, $\mathbf{W} = \{W_1, ..., W_L\}$, assigned to its layers. Given a batch of training samples $\mathbf{X}$ and the corresponding ground truth, the error of the network's error $\mathcal{L}$ can be defined as the difference between the predicted output $\mathbf{Y}$ and the ground truth. $\mathcal{L}$ is usually defined as cross entropy loss or mean squared error, *etc*. Training the deep neural networks involves solving an Empirical Risk Minimization problem.

Both the training and inference of deep neural networks rely on floating-point parameters, wherein weights and activations are stored with 32-bit precision. To reduce the computation and memory costs, model quantization methods represent the weights or activations in neural networks with low-bit values. Achieving this requires restricting floating-point numbers of weights $W_l$ and activations $A_l$ to a finite set of values. This is done using a quantization function, which is typically defined as:

$$\mathcal{Q}(z) = \Theta_i, \quad \forall z \in (t_i, t_{i+1}] \tag{1}$$

where $(t_i, t_{i+1}]$ denotes a real number interval $(i = 1, \cdots, 2^b)$, $b$ is the quantization bit-width, and $z$ is the input, *i.e.* weight or activation. The quantization function in Eq. equation 1 maps all the values in the range of $(t_i, t_{i+1}]$ to $\Theta_i$.

For ternary quantization, the number of intervals is set to three. Generally, ternary quantization methods can be distinguished into three categories: symmetric thresholds and scaling factors, symmetric thresholds and asymmetric scaling factors, asymmetric thresholds and symmetric scaling factors. To simplify the implementation, both the order and the sign of the network's parameters are usually preserved.

**Symmetric thresholds and scaling factors**    In the work of Li et al. (2016), the authors use two symmetric thresholds $\pm\Delta_l$ and symmetric scaling factor $\pm\alpha_l$ for the $l^{th}$ layer to quantize weights to $\{-\alpha_l, 0, +\alpha_l\}$:

$$\widehat{W_l} = \mathcal{T}(W_l|\Delta_l) = \begin{cases} \alpha_l, & \text{if } W_l > \Delta_l \\ 0, & \text{if } |W_l| \leq \Delta_l \\ -\alpha_l. & \text{if } W_l < -\Delta_l \end{cases} \tag{2}$$

in which $\widehat{W_l}$ represents the ternary weights and $\Delta_l$ is an positive threshold parameter. Then they solve an optimization problem of minimizing the $L_2$ distance between full precision and ternary weights to obtain layer-wise values of $\alpha_l$ and $\Delta_l$:

$$\begin{aligned} \Delta_l &= 0.7 \times \mathbf{E}(|W_l|), \\ \alpha_l &= \mathop{\mathbf{E}}_{i \in \{i | w_l(i)| > \Delta\}} (|W_l(i)|). \end{aligned} \tag{3}$$

**Symmetric thresholds and asymmetric scaling factors**    In the work of Zhu et al. (2017), the authors introduced two quantization factors $\alpha_l^p$ and $\alpha_l^n$ for positive and negative weights in $l^{th}$ layer. During feed-forward, quantized ternary weights $W_l$ are calculated as:

$$\widehat{W_l} = \mathcal{T}(W_l|\Delta_l) = \begin{cases} \alpha_l^p, & \text{if } W_l > \Delta_l \\ 0, & \text{if } |W_l| \leq \Delta_l \\ \alpha_l^n. & \text{if } W_l < -\Delta_l \end{cases} \tag{4}$$

in which two scaling coefficients $\alpha_l^p$ and $\alpha_l^n$ are two independent parameters and are trained together with other parameters. Following the rule of gradient descent, derivatives of $\alpha_l^p$ and $\alpha_l^n$ are calculated as:

$$\begin{aligned} \frac{\partial \mathcal{L}}{\partial \alpha_l^p} &= \sum_{i \in I_l^p} \frac{\partial \mathcal{L}}{\partial \widehat{W_l}(i)}, \\ \frac{\partial \mathcal{L}}{\partial \alpha_l^n} &= \sum_{i \in I_l^n} \frac{\partial \mathcal{L}}{\partial \widehat{W_l}(i)}. \end{aligned} \tag{5}$$

**Asymmetric thresholds and symmetric scaling factors**    In the work of Alemdar et al. (2017), the authors ternarize the weights of the $l^{th}$ layer using two thresholds $\Delta_l^p$ and $\Delta_l^n$ such that $\max(W_l) \geq \Delta_l^p \geq 0$ and $0 \geq \Delta_l^n \geq \min(W_l)$. The ternary weights for the $l^{th}$ layer are obtained by weight ternarization as follow:

$$\widehat{W_l} = \mathcal{T}(W_l|\Delta_l) = \begin{cases} \alpha_l, & \text{if } W_l > \Delta_l^p \\ 0, & \text{if } \Delta_l^p \geq W_l \geq \Delta_l^n \\ -\alpha_l. & \text{if } W_l < \Delta_l^n \end{cases} \tag{6}$$

However, the asymmetric thresholds can be be optimized by the normal STE (Straight-Through Estimator) and Alemdar et al. (2017) use a greedy search strategy with a manual setting score, which make the performance unsatisfactory.

### 3.2    BI-STE FOR ASYMMETRIC THRESHOLDS

We opted to use the third quantization way since the asymmetric thresholds can be easily implemented by simple comparison operation and the symmetric scaling factor is well-suited to matrix

multiplication. Previous quantization methods usually utilize Straight-Through Estimator (STE) to approximate the backward gradients in the quantization functions (Choi et al., 2018; Zhang et al., 2018; Esser et al., 2019). The core idea of STE is that when the weights are initialized, they are full-precision values like floating-point. During the forward pass, the weights are quantized to integer values and used for calculation, so that the output of quantized network can be calculated. Then the original full-precision weights rather than the quantized weights are updated during the backward propagation. The update of the entire network is completed during this process.

This simple approximation function works well for symmetric quantizer. However, STE implicitly enforces the equal axis aspect ratio in the input and output intervals of the quantizer because it regards the quantization function as an identity function in the backward propagation. Therefore, in this paper, we proposed a bi-STE optimization method for asymmetric thresholds.

Inspired by the stochastic theory, the stochastic ternarization can be formulated as:

$$
\widehat{z_l^i} = \begin{cases} +\alpha_l, & \text{with prob. } p = \text{clip}(\frac{z_l^i}{2\Delta_l^p}, 0, 1) \\ -\alpha_l, & \text{with prob. } p = \text{clip}(-\frac{z_l^i}{2\Delta_l^n}, 0, 1) \\ 0. & \text{others} \end{cases} \tag{7}
$$

where $\widehat{z_l^i}$ denotes the stochastic ternarized variables. The full-precision variables are quantized to $\{+\alpha_l, 0, -\alpha_l\}$ stochastically according to their distances to $\{+\alpha_l, 0, -\alpha_l\}$.

Correspondingly, the deterministic ternary function in the forward pass can be attained via setting a hard threshold on the probability (*i.e.*, $p = 0.5$) in Eq. 7 :

$$
\widehat{z_l^i} = \begin{cases} +\alpha_l, & \text{if } p_{\{\widehat{z_l^i}=1\}} \geq 0.5 \\ -\alpha_l, & \text{if } p_{\{\widehat{z_l^i}=-1\}} \geq 0.5 \\ 0. & \text{others} \end{cases} = \begin{cases} +\alpha_l, & \text{if } z_l^i \geq \Delta_l^p \\ -\alpha_l, & \text{if } z_l^i \leq \Delta_l^n \\ 0. & \text{others} \end{cases} \tag{8}
$$

### 3.3 UNIFORM DISTRIBUTION

From the perspective of information theory, more information are preserved when quantized parameters contain higher entropy. Based on the Lagrange multiplier, we can easily obtain the optimal state when the proportions of full-precision weights being quantized to multiple quantization levels are equal, and the information entropy in the quantized weights reaches its maximum value. Assume that the weights follow a Gaussian distribution, we empirically solve that when full-precision parameters are regularized to:

$$
\widetilde{Z}_l = \gamma_l \cdot (Z_l - \mu_l) \cdot \frac{0.5}{0.43 \cdot \sigma_l} Z_l, \tag{9}
$$

where $Z_l$ and $\widetilde{Z}_l$ are the full-precision and normalized weights/activations, $\mu_l$ and $\sigma_l$ denote the mean and standard deviation of the weights/activations in the $l^{th}$ layer. Since the entries in $[-0.43\sigma, 0.43\sigma]$ occupy about $1/3$ of the whole entries in Gaussian distribution, the corresponding parameters will be approximately uniformly quantized after the regularization.

Considering that the thresholds are asymmetric, a regularization factor $\gamma$ is employed to further adjust the distribution of the full-precision parameters, which is initialized as 1. After training, this regularization factor can be calculated offline from the optimized weights/activations and be absorbed by the BatchNorm layers after the quantized layers.

It should be noted that the distribution of the full-precision activation could be half of the Gaussian distribution, in which case, the regularization formulation can be altered accordingly.

### 3.4 TWO-STAGE TRAINING STRATEGY

Following the successful experience in prior binary methods (Liu et al., 2018; 2020; Martinez et al., 2020), a two-stage training strategy is utilized in our method. In the first training stage, a full-precision large model is used as the teacher model and the target model with ternary activations is

regarded as the student model. The knowledge distillation loss is employed to facilitate the training of the student network:

$$\mathcal{L} = \mathcal{L}_{\text{kd}}(\mathbf{y}_\text{s}, \mathbf{y}_\text{t}), \tag{10}$$

in which $\mathcal{L}_{\text{kd}}$ represents the knowledge distillation loss, $\mathbf{y}_\text{s}$ and $\mathbf{y}_\text{t}$ are the output probability of the student and teacher model.

For the second training stage, the same full-precision model is used as the teacher model and the target model with both ternary weights and activations is used as the student model. The loss function of the second stage is formulated as follows:

$$\mathcal{L}_{\text{total}} = \lambda\mathcal{L}_{\text{kd}}(\mathbf{y}_\text{s}, \mathbf{y}_\text{t}) + (1 - \lambda)\sum_{i=1}^{n}\mathcal{L}_{\text{ce}}(\mathbf{y}_\text{s}, \mathbf{y}_{\text{gt}}), \tag{11}$$

where $\mathcal{L}_{\text{ce}}$ is the cross-entropy loss, $\mathbf{y}_{\text{gt}}$ is the one-hot ground-truth probability and $\lambda$ is the trade-off hyper-parameter. It is worth noting that the corresponding weights of the second stage are initialized from the training results of the first stage.

## 4 Experimental Results

### 4.1 Training Details

**Datasets** The experiments are conducted on two benchmark classification dataset: CIFAR-10 and ISLVRC 2012 (simplified as ImageNet below) (Russakovsky et al., 2015). The CIFAR-10 dataset contains $50K$ training images and $10K$ test images with image size $32 \times 32$. We follow the standard data augmentation during training time, *i.e.*, padding 4 pixels on each side of an image and randomly flip it horizontally. The single view of the original image without padding or cropping is evaluated during test time. Top-1 accuracy is evaluated on CIFAR-10 dataset.

ImageNet is a large-scale image classification dataset with 1,000 classes, consisting of 1.28 million training images and $50K$ validation images. We use the standard scale and aspect ratio augmentation strategy from He et al. (2016) during training time. During test time, images are resized so that the shorter side is set to 256 and then cropped to $224 \times 224$. Top-1 and Top-5 classification accuracy are reported on ImageNet dataset.

**Implementation Details** Following the common setting in the quantization method, the ternary model is initialized with the full-precision pre-trained model. All the layers except the first and last layers are ternarized as conducted in the prior methods (Choi et al., 2018; Zhang et al., 2018; Esser et al., 2019). The RPReLU (Liu et al., 2020) structure is adopted as the activation layer which has been proved to be beneficial in binary neural networks. For hyper-parameters, the $\lambda$ for knowledge distillation is set to 0.9 in all the experiments, and $\gamma$ is set to 1 for all the layers which would be learned with the main network during the training process. It it worth noting that only one-stage training is utilized on CIFAR-10 dataset since the task of CIFAR-10 dataset is relatively simple.

### 4.2 Results on CIFAR-10

On CIFAR-10 dataset, two models are evaluated with the proposed method: ResNet-20 (He et al., 2016) and VGG-S (Zhu et al., 2017; Deng & Zhang, 2022). For ResNet-20, the initial learning rate is set to 0.1 and is divided by 10 at the $100^{th}$ and $150^{th}$ epoch, the model is trained for 200 epochs with mini-batch size 128. For VGG-S, the learning rate is set to 0.01 and is divides by 10 at the $150^{th}$ and $200^{th}$ epoch, where the model is trained for 300 epochs with mini-batch size 128. The weight decay and momentum are set to 1e-4 and 0.9 respectively for both ResNet-20 and VGG-S. SGD optimizer is used for CIFAR-10 dataset and ResNet-110 is utilized as the teacher model if needed.

The experimental results compared with existing ternary and low-bit quantization methods are shown in 1. For ResNet-20, ternary weights and full-precision methods such as TTQ (Zhu et al., 2017), BNN+(Darabi et al., 2018), LR-Net (Shayer et al., 2018), SCA (Deng & Zhang, 2022), surfer the drop of the accuracy. PACT (Choi et al., 2018) is a low-bit training-aware method which propose a parameterized clipping activation function, it achieves only 89.7% with 2-bit quantization.

Table 1: Comparison with ternary quantization and low-bit quantization methods on CIFAR-10 dataset. 'FP' represents the Full-Precision model and '*' denotes the models with knowledge distillation.

| Model | Method | W-Type | A-Type | FP Acc. | Top-1 Acc. | Acc. GAP |
|---|---|---|---|---|---|---|
| ResNet-20 | TTQ (Zhu et al., 2017) | Ternary | FP | 91.77% | 91.13% | -0.64% |
| | BNN+ (Darabi et al., 2018) | Ternary | FP | 91.77% | 90.10% | -0.67% |
| | LR-Net (Shayer et al., 2018) | Ternary | FP | 91.77% | 90.08% | -0.69% |
| | SCA (Deng & Zhang, 2022) | Ternary | FP | 91.77% | 91.28% | -0.49% |
| | PACT (Choi et al., 2018) | 2-bit | 2-bit | 91.60% | 89.70% | -1.90% |
| | LQ-Nets (Zhang et al., 2018) | 2-bit | FP | 92.10% | 91.80% | -0.30% |
| | ours | Ternary | Ternary | 91.78% | **92.35%** | +0.57% |
| | ours* | Ternary | Ternary | 91.78% | **92.97%** | +1.19% |
| VGG-S | TWN (Li et al., 2016) | Ternary | FP | 92.88% | 92.56% | -0.32% |
| | BNN+ (Darabi et al., 2018) | Ternary | FP | 93.37% | 90.32% | -0.05% |
| | LR-Net (Shayer et al., 2018) | Ternary | FP | 93.37% | 93.26% | -0.11% |
| | SCA (Deng & Zhang, 2022) | Ternary | FP | 93.42% | 93.41% | -0.01% |
| | TSQ (Wang et al., 2018) | Ternary | 2-bit | 93.42% | 93.49% | +0.07% |
| | LQ-Nets (Zhang et al., 2018) | 2-bit | FP | 93.80% | 93.80% | 0.00% |
| | LQ-Nets (Zhang et al., 2018) | 2-bit | 2-bit | 93.80% | 93.50% | -0.30% |
| | ours | Ternary | Ternary | 94.01% | **94.07%** | +0.06% |
| | ours* | Ternary | Ternary | 94.01% | **94.14%** | +0.13% |

LQ-Nets (Zhang et al., 2018) is a non-uniform quantization method, the performance of its 2-bit weight quantized model is still 0.30% lower than the full-precision model. As a comparison, the Top-1 accuracy of the proposed method is 0.57% better than that of the FP model without the help of knowledge distillation. The knowledge distillation can further boost the performance of proposed method to 92.97%.

For the VGG-S, it has more parameters than ResNet-20 and the performance of FP model is also better than that of ResNet-20. As shown in 1, SCA (Deng & Zhang, 2022) and LQ-Nets (Zhang et al., 2018) achieve the best results among the compared methods, where they are lossless with ternary and 2-bit weight respectively. When LQ-Nets quantize both weights and activations on VGG-S, it has a 0.3% Top-1 accuracy loss. The proposed method achieves the same accuracy compared with the full-precision model while ternarizing both weights and activations. And the performance can be even better than the full-precision model, which is 94.07%. We think that is because there are redundant parameters in VGG-S and the proposed method realizes an appropriate regularization for the model.

## 4.3 RESULTS ON IMAGENET

Three models are adopted to evaluate the proposed method on ImageNet dataset: ResNet-18, ResNet-34 and ResNet-50. The initial learning rate is set to 0.005 and the cosine learning rate decay scheduler is adopted. The model is trained using Adam optimizer with mini-batch size of 256. The weight decay is set to 0 and the model is trained for 100 epochs on each stage.

The experimental results compared with existing methods are shown in 2. For ResNet-18, the top-1 accuracy loss of TWN (Li et al., 2016), INQ (Zhou et al., 2017), ELQ (Zhou et al., 2018), TTQ (Zhu et al., 2017), SCA (Deng & Zhang, 2022) are 4.3%, 3.6%, 2.3%, 3.0% and 1.7% respectively. When the weights and activations are both quantized, the performance

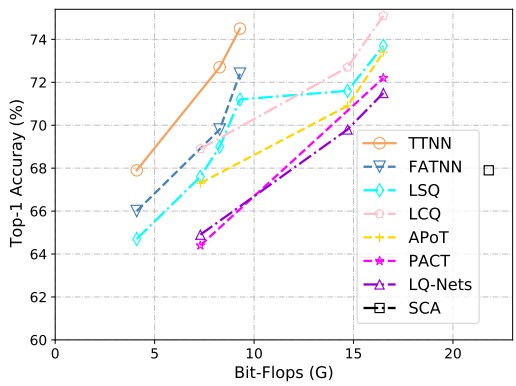

Figure 2: Comparison with existing ternary and low-bit quantization methods.

Table 2: Comparison with ternary and low-bit quantization methods on ImageNet dataset. '*' denotes the models with knowledge distillation.

| Model | Method | W-Type | A-Type | FP Acc. | Top-1 Acc. | Top-5 Acc. | Acc. GAP |
|---|---|---|---|---|---|---|---|
| ResNet-18 | TWN (Li et al., 2016) | Ternary | FP | 69.6% | 65.3% | 86.2% | -4.3% |
| | ELQ (Zhou et al., 2018) | Ternary | FP | 69.6% | 67.3% | 88.0% | -2.3% |
| | INQ (Zhou et al., 2017) | Ternary | FP | 69.6% | 66.0% | 87.1% | -3.6% |
| | TTQ (Zhu et al., 2017) | Ternary | FP | 69.6% | 66.6% | 87.2% | -3.0% |
| | SCA (Deng & Zhang, 2022) | Ternary | FP | 69.5% | **67.9%** | 88.0% | -1.6% |
| | LQ-Nets (Zhang et al., 2018) | 2-bit | 2-bit | 70.3% | 64.9% | 85.9% | -5.4% |
| | PACT (Choi et al., 2018) | 2-bit | 2-bit | 70.2% | 64.4% | – | -5.8% |
| | APoT (Li et al., 2020) | 2-bit | 2-bit | 70.5% | 67.3% | 87.5% | -3.2% |
| | LSQ (Esser et al., 2019) | 2-bit | 2-bit | 70.5% | 67.6% | – | -2.9% |
| | LSQ (Esser et al., 2019) | Ternary | Ternary | 69.8% | 64.7% | 85.6% | -5.1% |
| | FATNN (Chen et al., 2021) | Ternary | Ternary | 69.8% | 66.0% | 86.4% | -3.8% |
| | ours | Ternary | Ternary | 69.8% | **67.2%** | 87.0% | -2.7% |
| | ours* | Ternary | Ternary | 69.8% | **67.9%** | 87.9% | -1.9% |
| ResNet-34 | LQ-Nets (Zhang et al., 2018) | 2-bit | 2-bit | 73.8% | 69.8% | 89.1% | -4.0% |
| | LSQ (Esser et al., 2019) | 2-bit | 2-bit | 74.1% | 71.6% | – | -2.5% |
| | APoT (Li et al., 2020) | 2-bit | 2-bit | 74.1% | 70.9% | 89.7% | -3.2% |
| | LCQ (Yamamoto, 2021) | 2-bit | 2-bit | 74.1% | **72.7%** | – | -1.4% |
| | LSQ (Esser et al., 2019) | Ternary | Ternary | 73.3% | 69.0% | 88.8% | -4.3% |
| | FATNN (Chen et al., 2021) | Ternary | Ternary | 73.3% | 69.8% | 89.1% | -3.5% |
| | ours | Ternary | Ternary | 73.3% | **71.2%** | 89.9% | 2.1% |
| | ours* | Ternary | Ternary | 73.3% | **72.7%** | 90.7% | -0.6% |
| ResNet-50 | LQ-Nets (Zhang et al., 2018) | 2-bit | 2-bit | 76.4% | 71.5% | 90.3% | -4.9% |
| | PACT (Choi et al., 2018) | 2-bit | 2-bit | 76.9% | 72.2% | – | -4.7% |
| | APoT (Li et al., 2020) | 2-bit | 2-bit | 76.9% | 73.4% | 91.4% | -3.5% |
| | LSQ (Esser et al., 2019) | 2-bit | 2-bit | 76.9% | 73.7% | – | -3.2% |
| | LCQ (Yamamoto, 2021) | 2-bit | 2-bit | 76.8% | **75.1%** | – | -1.7% |
| | LSQ (Esser et al., 2019) | Ternary | Ternary | 76.1% | 71.2% | 90.1% | -4.9% |
| | FATNN (Chen et al., 2021) | Ternary | Ternary | 76.1% | 72.4% | 90.6% | -3.7% |
| | ours | Ternary | Ternary | 76.1% | **73.6%** | 91.0% | -2.5% |
| | ours* | Ternary | Ternary | 76.1% | **74.5%** | 91.6% | -1.6% |

drops more for 2-bit quantization. For example, APoT (Li et al., 2020) is a classical non-uniform quantization method which constrains all quantization levels as the sum of Powers-of-Two terms and LSQ (Esser et al., 2019) is a classical uniform quantization method which rescale the loss gradient of the quantization step size based on layer size and precision. Their top-1 accuracy of 2-bit models are 67.3% and 67.6% respectively. FATNN (Chen et al., 2021) design a novel ternary inner product with fully bit operations and ternarize both weights and activations. However, the top-1 accuracy loss of ternary LSQ and FATNN are 5.1% and 3.8% respectively. The proposed method achieves the best performance 67.9% and the least accuracy loss over them with both ternary weights and activations, which demonstrate the effectiveness of the proposed method.

For ResNet-34 and ResNet-50 model, we also compare the proposed method with low-bit and ternary quantization method. As we can see, the proposed method achieves a better result than LQ-Nets, APoT, LSQ and a similar result with LCQ base on a worse full-precision model on ResNet-34. Compared with ternary quantization method LSQ and FATNN, our method outperforms by 3.7% and 2.9% respectively, which is a large margin. On a much larger model ResNet-50, our method still outperforms most of the 2-bit quantization models and all the ternary methods.

## 4.4 ABLATION STUDIES

**Uniform Distribution and Asymmetric Thresholds** Firstly, we have conducted ablation study on the proposed uniform distribution and asymmetric thresholds. The experiments are accomplished with ResNet-18 model on ImageNet dataset and only one-stage training strategy is used for simpli-

fication. As shown in 3, the top-1 accuracy is only 65.2% when using symmetric thresholds and not using uniform normalization. Uniform normalization and asymmetric thresholds can improve the performance of ternary model by 1.0% and 1.1% respectively. Our method can achieve 67.5% top-1 accuracy when using both uniform normalization and asymmetric thresholds, which demonstrate the effectiveness of the proposed modules.

Table 3: Ablation study of uniform distribution and asymmetric thresholds.

| Uniform | Asymmetric | Top-1 Acc. | Top-5 Acc. |
|---------|------------|------------|------------|
| ✗ | ✗ | 65.2% | 85.8% |
| ✓ | ✗ | 66.2% | 86.0% |
| ✗ | ✓ | 66.3% | 86.3% |
| ✓ | ✓ | 67.2% | 87.0% |

**Two-stage Training Strategy**   Secondly, we have compared two-stage training with one-stage training strategy which learns ternary weights and activations jointly and is trained by the same epochs as the sum of two-stage training strategy. The experiments are completed with ResNet-18, ResNet-34 and ResNet-50 models on ImageNet dataset. As shown in 4.4, the performances of two-stage training strategy are always better than that of one-stage training. It is interesting that the gap between them are getting larger with the increase of the number of layers in the model. We consider that is because the ternary model is harder to train with the increase of the model parameters and two-stage training can fully exploit the representation ability of the parameters.

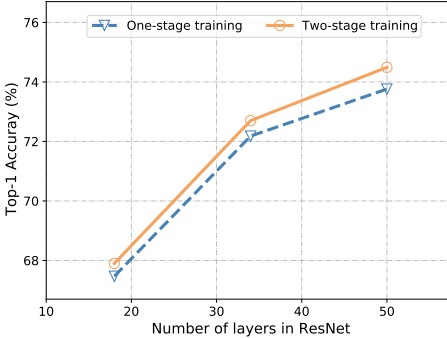
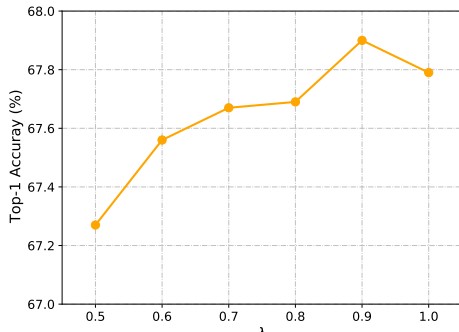

Figure 3: Ablation studies on two-stage training strategy and trade-off parameter for KD.

**Trade-off Parameter for KD**   Finally, we have tested the trade-off parameter $\lambda$ for knowledge distillation in the second stage. The experiments are conducted with ResNet-18 model on ImageNet dataset. As shown in 4.4, the top-1 accuracy of ternary model first increases and then decreases with the increase of $\lambda$ from 0.5 to 1.0. When the logits of teacher model and target plays the same role, the top-1 accuracy is only 67.3% and in contrast the top-1 accuracy is 67.8% when only use the logits of teacher model. Thus, we empirically set the $\lambda$ to 0.9. We consider the reason of this phenomenon is that the logits of teacher model is easier for student model to learn and that is also the point why the $\lambda$ is set to 1.0 in the first stage.

## 5   CONCLUSION

In this paper, we introduce a novel ternary quantization method that normalizes the weights and activations to follow a a uniform distribution with an underlying assumption of Gaussian distribution. We leverage asymmetric thresholds and symmetric scaling factors to maximize the representation capacity of TNN and the bi-STE optimization method is introduced. In addition, we employ a two-stage training strategy combined with the knowledge distillation method to enhance the performance of our proposed approach. Experimental results demonstrate the effectiveness of our method, with the aim of advancing the application of TNN. In the future, we plan to explore the co-design of the ternary algorithms with the ternary logic circuits.

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
