# OpenReview forum: "Revisiting Ternary Neural Networks towards Asymmetric Thresholds and Uniform Distribution"
_ICLR.cc/2024/Conference — Submitted to ICLR 2024_

### Official Review · Reviewer_FfEp · 2023-10-30

**Soundness:** 2 fair
**Presentation:** 1 poor
**Contribution:** 2 fair
**Rating:** 5
**Confidence:** 4

**Summary:**

This paper introduces a novel approach to enhance the performance of Ternary Neural Networks (TNNs). By leveraging a uniform distribution of quantized values and adopting an asymmetric thresholds and symmetric scaling factors scheme, the proposed method improves the representation ability of TNNs. Additionally, a two-stage knowledge distillation scheme is employed to further enhance performance. Experimental results demonstrate the effectiveness of the proposed method, achieving a top-1 accuracy of 74.5% for ResNet-50 on ImageNet, surpassing previous ternary quantization methods and even outperforming representative 2-bit quantization methods. Overall, this work presents a valuable contribution to the field of TNNs and their application in high-performance neural networks.

**Strengths:**

- The experimental results for both weights and activation quantized ternary networks are promising. It is encouraging to observe that ternary networks have achieved better results compared to 2-bit networks.
- Although there are some concerns regarding the implementation details, the proposed method appears to be straightforward and easy to understand.

**Weaknesses:**

# Major Concern:
- The explanation of how the asymmetric thresholds are optimized using the normal STE (Straight-Through Estimator) is not entirely clear. It would be beneficial for the authors to provide a more detailed derivation of the gradients for $\Delta_l^p$ and $\Delta_l^n$ and elaborate on the formulation of the bi-STE method. Additionally, it would be helpful to know whether the authors utilized stochastic ternarization or deterministic ternary function in their implementation.
- While the normalization of the distribution of Gaussian-like weights and activations using $\mu$ and $\sigma$, and the adjustment of variance with $\gamma$ seems reasonable, the additional multiplication by $Z_l$ appears somewhat unconventional. It would be valuable to understand if this technique was also applied to activation quantization during inference, and if so, it may introduce some additional computational costs.
- The introduction of the uniform distribution raises some questions about its necessity. Although the authors may alter the distribution of the original Gaussian through Equation (9), it should be noted that the product of two Gaussian random variables does not result in a uniform distribution. It may be worth considering modifying the threshold in Equation (8) to achieve the desired "approximately uniform quantization" of the corresponding parameters after regularization.

# Minor Issue:
- The authors mentioned the utilization of RPReLU [1] to enhance the results, but it would be beneficial to have an ablation study to evaluate the impact of this setting. Additionally, comparing the results of two-stage training with other low-bit quantization methods may not provide a fair comparison. It is suggested to include an additional column in Table 1 and Table 2 to compare the training costs of different methods.

# Reference:
- [1] Reactnet: Towards precise binary neural network with generalized activation functions. ECCV2020

**Questions:**

- The authors state that "Studies have shown that the cost factor for ternary circuits is close to 3" and "In binary logic circuits, resulting in $cost=2$." However, this seems inconsistent with the conclusion that "the actual efficiency of ternary systems surpasses that of binary systems."
- It is unclear if $b$ is involved in Equation (1) since it is not explicitly mentioned. Additionally, the quantization function is defined on the real number interval. Did the authors imply that a clip function was applied before quantization?
- The absence of asymmetric thresholds and scaling factors in ternary quantization raises curiosity. It would be helpful if the authors could provide an explanation for this.
- In Table 2, for the results of the proposed method without knowledge distillation, it is unclear if the two-stage training strategy was still employed.
- Further clarification is needed regarding the statement "After training, this regularization factor can be calculated offline from the optimized weights/activations." It would be beneficial if the authors could elaborate on how the regularization factor is recalculated after training.
- According to the statement "It should be noted that the distribution of the full-precision activation could be half of the Gaussian distribution, in which case, the regularization formulation can be altered accordingly," it appears that unsigned quantization is used for activation, i.e., $\\{0, +1, +2\\}$. If this is the case, the activation quantization seems inconsistent with the bi-STE settings. It would be helpful if the authors could provide further explanation on this matter.
- Since the proposed "TWO-STAGE TRAINING STRATEGY" significantly overlaps with existing works, it may be more appropriate to refrain from claiming it as a contribution of this work and omit it from the main body of the paper.

---

### Official Review · Reviewer_8ZRJ · 2023-10-31

**Soundness:** 2 fair
**Presentation:** 2 fair
**Contribution:** 1 poor
**Rating:** 3
**Confidence:** 5

**Summary:**

This paper explores the techniques to improve the training of Ternary Neural Networks (TNNs). More precisely, the paper introduces asymmetric thresholds, 2-stage knowledge distillation (KD) and regularization techniques. The methods are validated using ResNet models on CIFAR and ImageNet datasets.

**Strengths:**

1). The paper is reasonably well-written.

2). TNN is an important subject, especially for large-scale models and on resource constrained edge devices.

3). The results demonstrate enhanced performance on ResNet Imagenet compared to selected methods.

**Weaknesses:**

1). The primary concern is the extreme lack of novelty. Nearlyall the techniques introduced in the paper have been previously proposed in existing literatures, including the learnable asymmetric threshold, 2-stage KD and entropy-based regularization. In fact, the first reference in the paper introduced the asymmetric threshold and 2-stage KD, but the author only very briefly mentioned in the main context.

2). Evaluation is limited in scope.  The evaluation is primarily done on ResNet model, which, while important, is somewhat outdated. It is expected that more state-of-the-art CNN and transformer models should be assessed to demonstrate the efficacy of these techniques.

3). The performance improvement is limited. Even with combined existing techniques and the utilization of optimized process (such as using RPReLU), the performance improvement is limited. Notably, the TNN ResNet shows worse results than  2-bit version, which has similar computational complexity in hardware implementation.

**Questions:**

Not clear to me the arguments on STE and proposed stochastic rounding approach. The authors argue that “STE implicitly enforces the equal axis aspect ratio in the input and output intervals”, however, why can the proposed stochastic rounding solve the problem?

---

### Official Review · Reviewer_TP94 · 2023-10-31

**Soundness:** 2 fair
**Presentation:** 3 good
**Contribution:** 2 fair
**Rating:** 5
**Confidence:** 4

**Summary:**

This paper introduces a sequence of techniques to train an accurate ternary neural network. These techniques include asymmetric quantization bins, making input values equally fall into quantization bins, and two-stage training (same as used in BNNs). The paper is motivated by the recent advances in ternary logic gates instead of binary. Experimental results show the proposed network accuracy is close to the 2-bit models.

**Strengths:**

1. The motivation of utilizing ternary logic gates provides new interesting lessons for quantization researchers.

2. The paper conducted good ablation studies.

3. The paper has a good presentation structure.

**Weaknesses:**

1. The paper needs to provide an efficiency analysis and more concrete description on the ternary system. For example, what will the storage format be? Are quantized weights and activations still stored in 2-bits or a special ternary format? Are matmul accumulations (supposed to be high-precision) stored in 32-bit binary format or a special ternary format? These discussions will affect the memory efficiency and change the view of where TNN is positioned between BNN and 2-bit NNs.

2. Without efficiency analysis, e.g., estimated performance or memory savings coming from the ternary logic gates, it is hard to differentiate between a real “ternary” network and a 2-bit network. The experimental results in the paper show that the accuracy of ternary networks is close to that of 2-bit ones. In some context of symmetric integer quantization, as some of the baselines are, a 2-bit network {-1, 0, +1} is actually ternary. Readers cannot precisely evaluate the benefits of the proposed ternary models over the previous 2-bit models.

**Questions:**

The paper mainly compares the proposed ternary model to previous ternary or 2-bit models. What would be the comparison to some of the recent 1-bit models, e.g., to [1] and [2]?

[1] N. Guo et al., Join the High Accuracy Club on ImageNet with A Binary Neural Network Ticket, arxiv’22.

[2] Y. Zhang et al., PokeBNN: A Binary Pursuit of Lightweight Accuracy, CVPR’22.

---

### Official Review · Reviewer_Dqjz · 2023-11-01

**Soundness:** 3 good
**Presentation:** 3 good
**Contribution:** 2 fair
**Rating:** 3
**Confidence:** 4

**Summary:**

In this paper, the authors tackled the quantization problem of ternary neural network. The optimization framework presented in this paper can enhance the representation capacity of ternary neural network and balance the distribution of quantization candidates. Besides, they
implemented a 2-stage knowledge distillation method to further enhance model performance.

**Strengths:**

1. This paper is easy to read and provided an overall background information of existing TNN
optimization methods for introducing their motivation.
2. This paper compared their proposed methods with existing solutions, including ternary
neural networks and binary neural networks, via experiments on different backbone models,
which demonstrated the performance of their TNN framework.
3. This paper leveraged the stochastic theory and Information theory to propose ternary
quantization methods whose performance (i.e., representation capacity) improvement possibly
could be explainable by mathematic proof.

**Weaknesses:**

1. The proposed methods proposed an optimization method named bi-STE based on stochastic
theory. But in equation 8, the deterministic ternary function in the forward pass set a hard
threshold on the probability and now it’s not stochastic. It will be helpful to explain how to
select hard thresholds that enable the variables in the model to be correctly quantized.
2. This paper argued that uniform distribution of weights/activations can enhance the
information capacity of the model. However, it only used a few experiments in Table 3 to show
it, which is not enough to support the conclusion. It would be better if authors could provide
more mathematical proof or experiment results to demonstrate it.
3. Some figures in this paper are not clear enough (e.g., figure 2), and hard to find the
corresponding line of your approach.
4. In the section 3.3, this paper assumes the weights follow a Gaussian distribution and
correspondingly proposes their method. Hence, it will be better if the method can be more
general and suitable for non-Gaussian distribution.

**Questions:**

1. Could you clarify how do you set the hard asymmetric thresholds for the TNN? And would the hard asymmetric thresholds infect the quantization performance?
2. In section 3.3, this paper assumes the weights follow a Gaussian distribution and correspondingly proposes their method. If some weights/activations layers don’t roughly follow the Gaussian distribution, would it seriously hurt your quantization performance?

---

### Meta-Review · Area_Chair_ffDw · 2023-12-05

**Metareview:**

Four experts reviewed the paper, and none was supportive. There was no rebuttal.

**Justification For Why Not Higher Score:**

No reviewer was supportive.

**Justification For Why Not Lower Score:**

No lower score available

---

### Decision · Program_Chairs · 2024-01-16

Reject